# Updated Sequence and Annotation of the Broad Host Range Rhizobial Symbiont *Sinorhizobium fredii* HH103 Genome

**DOI:** 10.3390/genes16091094

**Published:** 2025-09-16

**Authors:** Francisco Fuentes-Romero, Francisco-Javier López-Baena, José-María Vinardell, Sebastián Acosta-Jurado

**Affiliations:** Department of Microbiology, Faculty of Biology, University of Seville, 41012 Sevilla, Spain; ffuentesr@us.es (F.F.-R.); jlopez@us.es (F.-J.L.-B.); sacosta@us.es (S.A.-J.)

**Keywords:** *Sinorhizobium fredii*, strain HH103, *Sinorhizobium*, genome sequencing, PacBio, Illumina, symbiotic plasmid, transposable elements

## Abstract

**Background**: *Sinorhizobium fredii* HH103 is a fast-growing rhizobial strain capable of infecting a broad range of legumes, including plants forming determinate and indeterminate nodules, such as *Glycine max* (its natural host) and *Glycyrrhiza uralensis*, respectively. Previous studies reported the sequence and annotation of the genome of this strain (7.25 Mb), showing the most complex *S. fredii* genome sequenced to date. It comprises seven replicons: one chromosome and six plasmids. Among these plasmids, pSfHH103d, also known as the symbiotic plasmid pSymA, harbors most of the genes involved in symbiosis. Due to limitations of the sequencing technology used at the time and the presence of high number of clusters of transposable elements, this plasmid could only be partially assembled as four separated contigs. **Methods**: In this work, we have used a combination of PacBio and Illumina sequencing technologies to resolve these complex regions, obtaining an updated genome sequence (7.27 Mb). **Results**: This updated version includes an increase in size of the largest replicons (chromosome, pSfHH103d, and pSfHH103e) and a complete and closed symbiotic plasmid (pSfHH103d or pSymA). Additionally, we carried out a re-annotation of the updated genome, merging the previous annotation and the new one found in the remaining gaps. Notably, we found a high number of transposable elements in the HH103 genome, especially in three plasmids (pSfHH103b, pSfHH103c, and pSymA), a feature that is common among *S. fredii* strains. **Conclusions**: The combination of PacBio and Illumina sequencing technologies has allowed us to obtain a complete version of the HH103 pSymA. The presence of a high number of mobile elements seems to be a general characteristic among *S. fredii* strains, a fact that might be related to a high genome plasticity.

## 1. Introduction

*S. fredii* is a rhizobial species able to establish a nitrogen-fixing symbiosis with dozens of legumes [1,2,3,4]. Most *S. fredii* strains have been isolated from Chinese soils and are able to nodulate soybeans (*G. max*) and wild soybeans (*Glycine soja*), their natural hosts [2,5]. However, a small number of *S. fredii* strains have been isolated in other geographical locations. This is the case for strains NGR234, isolated from nodules of *Lablab purpureus* in New Guinea [1], and SMH12, isolated from soybean nodules in Vietnam [6,7]. Interestingly, NGR234 is unable to nodulate soybean, but it can induce the formation of nitrogen-fixing nodules in different *G. soja* accessions from Central China [8]. The rest of *S. fredii* strains nodulate effectively Asiatic varieties of soybeans, and many strains are also able to nodulate American (commercial) varieties of this legume [2,9]. At least in the well-studied cases, this symbiotic (in)compatibility is caused by plant recognition through resistance (R) proteins of effectors delivered into host cells by a symbiotic type 3 secretion system (T3SS) [10]. Among the best studied *S. fredii* strains, USDA257 and HH103 are the representative examples of strains nodulating only Asiatic soybeans and strains nodulating both Asiatic and American soybeans, respectively [2,10].

The genome of *S. fredii* HH103 was first sequenced in 2012 [11,12]. Its size was 7.25 Mb and consisted of a chromosome and 6 plasmids, being, to our knowledge, the most complex *S. fredii* genome analyzed so far (https://www.ncbi.nlm.nih.gov/datasets/genome/?taxon=380, accessed on 31 July 2025). The number of plasmids present in other *S. fredii* strains varies between one (strain USDA257) and four (Strain Sf1), with the presence of two megaplasmids being the most frequent case (for example, strains NGR234 and SMH12). An interesting study of the complexity of *S. fredii* genomes and the lineage-specific adaptations among soybean-nodulating rhizobia has been provided by Tian and collaborators [13]. Among HH103 plasmids, the symbiotic plasmid (plasmid d or pSymA), which harbors most of the genes involved in symbiosis, could only be partially assembled (4 contigs) due to limitations of the sequencing technology (454 Life Sciences) and the presence of a high number of clusters of repeated sequences [12].

In this work, we have used a combination of two sequencing technologies, PacBio and Illumina, to obtain a new genome sequence of HH103 that includes a complete version of the pSymA plasmid. The updated HH103 genome sequence (7.27 Mb) shows an increase in size of the largest replicons (chromosome, plasmid c, plasmid d and plasmid e) in comparison to the previous version of the genome, and the new detected genes have been manually annotated. Also, the annotation of the gene coding for the T3SS effector protein NopD has been updated. The HH103 genome harbors 6949 open reading frames (ORFs), including 340 related to transposable elements (110 located on the pSymA).

## 2. Materials and Methods

For DNA isolation, *S. fredii* HH103 was streaked from a glycerol stock conserved at −80 °C onto tryptone-yeast extract (TY) medium [14] and cultured at 28 °C for a week. Single colonies from the plate were inoculated into TY broth and grown for three days at 28 °C and 180 rpm. DNA from bacterial cultures was extracted using Monarch genomic DNA purification kits (New England Biolabs, Ipswich, MA, USA) following the protocol supplied by the manufacturer. Extracted DNA was sequenced at Novogene (Cambridge, UK) using Pacific Biosciences (PacBio, Menlo Park, CA, USA) Technology Sequel II (CLR mode). Illumina sequencing was performed at Macrogen using Novaseq 600 150PE (Seoul, Republic of Korea) to produce 2 × 150 bp reads. Illumina reads were assessed using FastQC v0.12.1 and no residual adapters or low-quality regions were identified, so no trimming or filtering steps were applied.

The assembly and draft genome of HH103 were generated from the PacBio reads using Flye v2.9.3-b1797 [15]. Contigs shorter than 3000 bp were discarded. Draft genome assembly improvement was carried out with Illumina reads using Pilon v1.24 [16]; read mapping for this step was performed with bowtie2 v2.5.3 [17]. To reorient replicons to start at the *dnaA* or *repA* gene, Circlator v1.5.5 was used with the fixstart options enabled [18]. Finally, the assembly was annotated by extracting the coordinates from the previous annotation and their sequence by using bedtools v2.31.1 [19] with the option getfasta. Those sequences were then searched in the new assembly by using the blastn command [20], and the coordinates of the matching regions were used to make a new annotation. The new genome was larger than the previous one, so the remaining gaps were annotated with bakta v1.11.0 [21] and this information was manually added to the new annotation file. Default settings and twenty-four threads, when multithreading was possible, were used for all the software employed in this work (unless otherwise specified). Sequencing statistics are provided in Table 1.

The automatic annotation provided by RefSeq (Locus_tag=“ACN6KE_RS…”) is available at https://www.ncbi.nlm.nih.gov/datasets/genome/GCF_048585425.1/ (accessed on 31 July 2025). The manual annotation generated in this work (Locus_tag=“ACN6KE_…” is available at https://github.com/ffuentesr97/08_25_HH103_genome.git (accessed on 1 September 2025).

A maximum likelihood phylogeny of 15 species from the genus *Sinorhizobium* was constructed as described previously [22]. *Rhizobium leguminosarum* SM52 was included as an outgroup.

Scripts to search mobile element genes in *S. fredii* strains and to plot the HH103 pSym are available at GitHub, software version 3.17.3 (https://github.com/ffuentesr97/08_25_HH103_genome.git, accessed on 1 September 2025). These scripts were run using the annotation and/or the nucleotide sequence, as indicated in the script.

The accession numbers of the other genomes analyzed in this work are as follows: *R. leguminosarum* SM52, GCF_004306555.1; *Sinorhizobium americanum* CCGM7, GCF_000705595.2; *Sinorhizobium kummerowiae* CCBAU71714, GCF_030064585.1; *Sinorhizobium medicae* WSM419, GCF_000017145.1; *Sinorhizobium meliloti* 1021, GCF_000006965.1; *S. meliloti* Rm41, GCF_002197045.1; *Sinorhizobium sojae* CCBAU05684, GCF_002288525.1; *Sinorhizobium terangae* CB3126, GCF_029714365.1; *S. fredii* CCBAU45436, GCF_003100575.1; *S. fredii* NGR234, GCF_000018545.1; *S. fredii* SMH12, GCF_024400375.1; *S. fredii* USDA192, GCF_041260365.1; *S. fredii* USDA193, GCF_041262265.1; *S. fredii* USDA205^T^, GCF_009601405.1; *S. fredii* USDA257, GCF_000265205.3.

## 3. Results and Discussion

### 3.1. Characteristics of the Updated Version of the S. fredii HH103 Genome

As described in the Materials and Methods Section, we have *de novo* sequenced the *S. fredii* HH103 genome by combining PacBio and Illumina technologies. This approach allowed us to obtain the pSymA sequence into a single contig (in the previous version, this sequence was split into 4 non-assembled contigs). As stated previously by our group [11,12], the HH103 genome consists of seven replicons: the chromosome and six plasmids (named as e, d, c, b, a2, a1), confirming that it is the most complex *S. fredii* genome sequenced so far. The GenBank accession numbers for the updated sequences of the seven replicons are as follows: CP183939.1 to CP183945.1. This new version of the HH103 genome is slightly bigger (7.27 Mb) than the previous one (7.25 Mb) since it entails an increase in size of the largest replicons (chromosome and plasmids d and e). Details about the gene content of each replicon have been previously described [12]. The largest plasmids, pSfHH103e and pSfHH103d, correspond to the pSymB and pSymA plasmids, respectively, described in *S. meliloti* and in other *S. fredii* strains [23,24]. Hereafter, we will refer to them as HH103 pSymB and pSymA, respectively. Thus, HH103 pSymB carries genes related to exopolysacharide production, one important rhizobial symbiotic signal [25], whereas HH103 pSymA harbors genes related to the production of Nod factors and the symbiotic T3SS as well as *nif* and *fix* genes required for nitrogen fixation inside nodules [12]. Table 2 contains the comparison between the previous and the updated versions of the HH103 genome, replicon by replicon, and Figure 1 shows a circular plot of HH103 pSymA (CP183945.1).

This complete sequence of the pSymA, which contains key symbiotic genes, should provide much better tools for researchers to probe the functions of these genes and further our understanding of legume-rhizobia symbiosis in a strain that is noteworthy for its broad host range. As we will comment below, the fully assembled HH103 pSymA plasmid contains a remarkably high number of mobile elements. It is noticeable that the updated version of the HH103 genome confirmed the presence of a second copy of the *nifHDK* genes (ACN6KE_004317 to ACN6KE_004319), also located (as the first ones: ACN6KE_004532 to ACN6KE_004530) on the pSymA (Figure 2). These two sets of copies of *nifHDK*, which are separated by around 190 kb in the sequence of pSymA, were 100% identical and preceded by a NifA-box, suggesting a similar regulation of their expression. Since the *nifHDK* genes code for the structural units of the nitrogenase [26], the presence of two copies of these genes might be related to nitrogen fixation efficiency inside nodules. The presence of two copies of *nifHDK* is common among *S. fredii* strains, in contrast to *S. meliloti*, where the common fact is the presence of a single copy of these genes. It is also relevant that the annotation of the *nopD* gene (ACN6KE_004365) has changed in the updated version of the HH103 genome. The *nopD* gene, also located on pSymA, codes for a T3SS effector protein [10,27] harboring a C48 protease domain that is involved in SUMOylation and de-SUMOylation of host proteins. The updated sequence of NopD is 1490 residues long, whereas the previous version had 1318 residues.

In addition to the presence of additional copies of previously described HH103 genes (especially mobile elements), we have found 16 new genes in the updated version of the genome (Table 3), most of them located on the chromosome, two in the pSymA, and one in the pSymB. Seven out of these 16 genes code for hypothetical proteins, whereas the rest encode highly diverse proteins.

### 3.2. Comparison of the Genome of HH103 with Other Rhizobia

We have compared the updated genome sequence of *S. fredii* HH103 with the previous one and with that of different *S. fredii* strains and *Sinorhizobium* species by using core-genome gene phylogeny. These comparisons gave rise to a phylogenetic tree, shown in Figure 3. As expected, *S. fredii* genomic sequences clustered together. Both HH103 genome sequences clustered with a set of *S. fredii* strains: USDA192, SMH12, USDA193, and USDA205. All these genomic sequences were closer to those of CCBAU45436 and NGR234, whereas the genome of USDA257 resulted in being the most different with respect to the rest of *S. fredii* strains. Regarding the comparisons between *S. fredii* strains and the other *Sinorhizobium* species included in this study, interestingly, genomes of *S. fredii* strains were more similar to that of *S. americanum* (an species that was first isolated from *Acacia* nodules in Mexico but that is also able to nodulate *Phaseolus*) [28,29] than to that of *S. sojae*, which was isolated from soybean nodules in China [30]. All the mentioned genomes are closer to *S. medicae*, *S. kummerowiae* and *S. meliloti* than to *S. terangae*. Our results are consistent with those obtained by Kuzmanović and collaborators [31] by using the same methodology.

### 3.3. The HH103 Genome Is the Most Complex Among the Different S. fredii Strains Characterized So Far

The updated version of the HH103 genome confirms its previously described complexity [12]. As mentioned above, this genome is composed of 1 chromosome and 6 different plasmids. Besides comparing the genome sequences of the different *S. fredii* strains analyzed in this work, we analyzed their genome structures. As shown in Table 4, the *S. fredii* genome sizes studied vary between 6.6 (strain USDA205T) and 7.3 Mb (strain HH103). Thus, among the different *S. fredii* strains analyzed, the genome of HH103 is not only the most complex but also the largest one. Except for USDA257, whose genome is described as composed of a chromosome and a single plasmid [32,33], all the strains harbor at least two plasmids, that, in all the strains in which this information is available, correspond to the two typical symbiotic megaplasmids (pSymB and SymA) of about 2 and 0.5 Mb also present in *S. meliloti* [12,23,24]. The single plasmid described in USDA257 corresponds to the megaplasmid that carries genes coding for Nod factor production and secretion, the symbiotic T3SS, and genes related to nitrogen fixation (pSymA) [32]. In conclusion, the *S. fredii* genome structure is highly variable among different strains.

### 3.4. S. fredii Strains Harbor Higher Numbers of Mobile Elements than Other Sinorhizobium Species

The *S. fredii* HH103 genome carries a high number of mobile elements. In fact, the high number of clusters of transposases and insertion sequences present in the pSymA prevented the full assembly of this replicon in the previous version of the HH103 genome [12]. Actually, the number of genes related to mobile elements present in the HH103 genome is 340. The presence of a high number of mobile elements in HH103 prompted us to investigate whether this is also the situation in other *S. fredii* strains and *Sinorhizobium* species. For this purpose, we carried out a search for mobile elements in the whole genomes of the rhizobial strains, and the results are shown in Table 5.

The number of genes related to mobile elements found in *S. fredii* strains was always higher than 200, oscillating between 209 and 356, present in USDA257 and USDA205, respectively. The genome of HH103 harbors 340 genes related to transposable elements. These numbers were higher than those found in other *Sinorhizobium* species, which varied between 108 in *S. terangae* CB3126 and 194 in *S. sojae* CCBAU05684. These data suggest a higher genome plasticity in *S. fredii* than in other *Sinorhizobium* species, since it is well known that mobile elements have an important impact on genome structure and function [34].

Finally, we decided to investigate the distribution of mobile elements among the different replicons of HH103. As shown in Table 6, the density of genes related to mobile elements (calculated as the number of these genes per 10 kb) varied enormously between the different replicons, being low (≤0.40) for the two largest replicons (chromosome and pSymB) and one of the two smallest plasmids (pSfHH103a2), medium (0.83) for pSfHH103a1, and high (>1.82) for plasmids pSfHH103b, pSfHH103c, and pSymA. Whether this higher presence of mobile elements in these three plasmids is related to a higher plasticity and/or a higher probability of gene horizontal transfer phenomena of these replicons in comparison with the rest of the HH103 genome remains to be investigated.

## 4. Conclusions

The use of two different technologies, one providing long-length fragments (PacBio) and the other resulting in a very high number of short reads (Illumina), has allowed us to solve the problem of the presence of a high number of clusters of transposable elements for the assembly of the pSymA plasmid. This approach might be adequate for the sequencing and assembling of other complex genomes. The *de novo* generated HH103 sequence confirmed that the HH103 genome, 1 chromosome and six plasmids, is the most complex among the 34 different *S. fredii* strains whose genomes have been sequenced so far (https://www.ncbi.nlm.nih.gov/datasets/genome/?taxon=380, accessed on 31 July 2025). The fact of having a complete sequence of the pSymA plasmid will facilitate further studies on *S. fredii* HH103 genes relevant for symbiosis. We have also shown that *S. fredii* genomes exhibit a higher presence of mobile elements than other *Sinorhizobium* species, a fact that might be related to a greater genomic plasticity and horizontal gene transfer probability. In fact, the role of insertion sequences in *S. fredii* strains’ adaptive evolution to symbiosis with their host plants has been previously proposed [35,36]. Also, a recent study highlights the importance of rhizobial mobile gene clusters in driving partner quality variation in symbiosis [37]. In the case of HH103, the presence of mobile elements is especially abundant in the pSymA, opening the possibility of the influence of those transposable elements in adaptation to the different partners of this strain.

## Figures and Tables

**Figure 1 genes-16-01094-f001:**
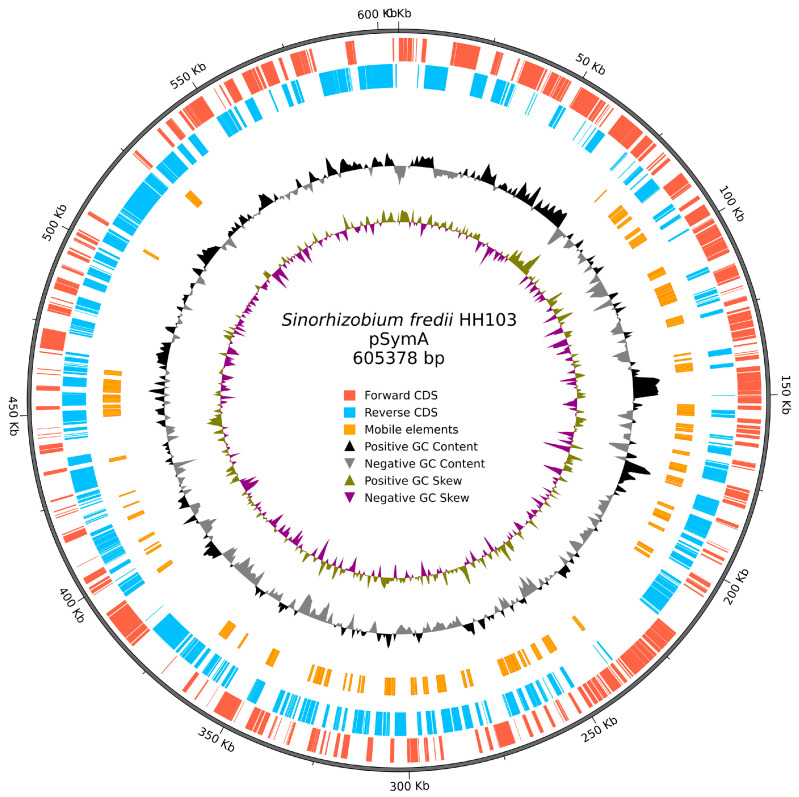
Circular plot of pSfHH103d (pSymA). The legends (see inside the plot) show what each circle represents.

**Figure 2 genes-16-01094-f002:**
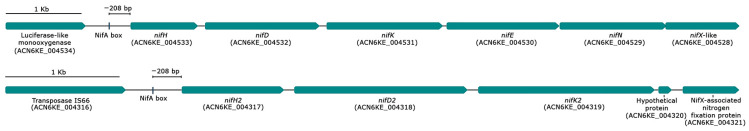
Genetic context of the two copies of *nifHDK* present in the *S. fredii* HH103 genome. The locus tags of the manual annotation are shown in brackets.

**Figure 3 genes-16-01094-f003:**
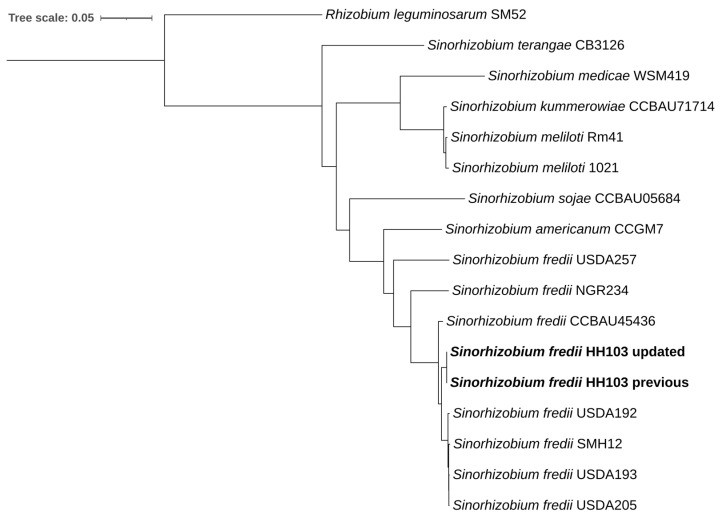
Maximum-likelihood core-genome gene phylogeny of different *Sinorhizobium* representative species and strains. *R. leguminosarum* SM52 was included as an outgroup. The scale represents the mean number of nucleotide substitutions per site.

**Table 1 genes-16-01094-t001:** Summary of sequencing statistics for the *S. fredii* HH103 genome.

BioProject accession no.	PRJNA1233244
BioSample accession no.	SAMN47263981
GenBank Assembly accession no.	GCF_048585425.1
GenBank accession no.	CP183939, CP183940, CP183941, CP183942, CP183943, CP183944, CP183945
SRA accession no.	-
PacBio reads	SRR34701253
Illumina reads	SRR34710307
Total PacBio read length (nt)	1,170,389,454
No. of PacBio reads	98,229
PacBio N_50_ read length (nt)	12,887
Total Illumina read length (nt)	1,782,256,322
No. of Illumina paired reads	11,803,022
Illumina read length (nt)	2 × 151
Genome size (bp)	7,273,959
No. of protein-coding genes	6949
G+C content (%)	62.14
Genome coverage	147×
No. of replicons	7
Replicon sizes (bp)	24,038; 25,081; 61,874; 144,081; 605,378; 2,099,565; 4,313,942

**Table 2 genes-16-01094-t002:** Comparison of the main characteristics of the previous and updated versions of the *S. fredii* HH103 genome.

Replicon		Chromosome	pSfHH103e (p_e, pSymB))	pSfHH103d (p_d, pSymA)	pSfHH103c (p_c)	pSfHH103b (p_b)	pSfHH103a2 (p_a2)	pSfHH103a1 (p_a1)
Length (bp)	Previous	4,305,723	2,096,125	ca. 588,797 ^a^	144,082	61,880	25,081	24,036
	Updated	4,313,942	2,099,565	605,378	144,081	61,874	25,081	24,038
GC content (%)	Previous	62.61	62.38	59.59	58.68	58.47	58.02	58.21
	Updated	62.60	62.38	59.56	58.68	58.47	58.03	58.21
CDS	Previous	4008	1991	664	169	62	38	19
	Updated	4013	1982	665	169	62	38	20
t-RNA genes	Previous	53	0	1	0	0	0	0
	Updated	53	0	1	0	0	0	0
*rrn* operons	Previous	3	0	0	0	0	0	0
	Updated	3	0	0	0	0	0	0
GenBank Accession number	Previous	HE616890	HE616899	CDSA010000001 to CDSA010000004	HE616893	HE616892	LN735562	HE616891
	Updated	CP183939.1	CP183944.1	CP183945.1	CP183943.1	CP183942.1	CP183941.1	CP183940.1

^a^ four concatenated contigs.

**Table 3 genes-16-01094-t003:** Genes previously not described in the *S. fredii* HH103 genome. The GeneID corresponds to the manual annotation performed in this work. The annotation generated automatically (RefSeq) is shown in brackets. Note that several genes are not detected by the automatic annotator.

Locus_Tag ^a^	Replicon	Description
ACN6KE_000391	Chromosome	Zinc metalloendopeptidase
ACN6KE_001521	Chromosome	Hypothetical protein
ACN6KE_001826	Chromosome	Antifreeze protein
ACN6KE_001897	Chromosome	ABC transporter ATP-binding protein
ACN6KE_001899	Chromosome	Hypothetical protein
ACN6KE_001932	Chromosome	Hypothetical protein
ACN6KE_002195	Chromosome	Hypothetical protein
ACN6KE_002415	Chromosome	RTX toxin hemolysin-type protein
ACN6KE_003412	Chromosome	IS21 family transposase
ACN6KE_003587	Chromosome	Peptidoglycan-binding protein LysM
ACN6KE_003588	Chromosome	Hypothetical protein
ACN6KE_003589	Chromosome	Hypothetical protein
ACN6KE_0035890	Chromosome	Imidazole glycerol phosphate synthase subunit HisF
ACN6KE_004165	pSymA	TIR domain-containing protein
ACN6KE_004181	pSymA	Hypothetical protein
ACN6KE_006263	pSymB	DUF1059 domain-containing protein

^a^ According to the manual annotation.

**Table 4 genes-16-01094-t004:** Comparison of the genome structure of several relevant *S. fredii* strains. When possible, the plasmid containing *nod* and *nif* genes is denoted in bold.

*S. fredii* Strain	Genome Accession Number ^a^	Genome Size (Mb)	Genome Structure (Sizes in Mb in Brackets)
CCBAU45436	GCF_003100575.1	6.9	1 chromosome (4.16), four plasmids: **a (0.42)**, b (1.96), d (0.20), e (0.17)
HH103	GCF_048585425.1	7.3	1 chromosome (4.31), six plasmids: a1 (0.024), a2 (0.025, b (0.062), c (0.14), **d (0.61)**, e (2.10)
NGR234	GCF_000018545.1	6.9	1 chromosome (3.92), two plasmids: **a (0.54)**, b (2.43)
SMH12	GCF_024400375.1	7.0	1 chromosome (4.02), two plasmids: **a (0.56)**, b (2.39)
USDA192	GCF_041260365.1	6.9	4 contigs, not fully assembled (1 chromosome, three plasmids)
USDA193	GCF_041262265.1	6.8	3 contigs, not fully assembled (1 chromosome, two plasmids)
USDA205^T^	GCF_001461695.1	6.6	209 contigs, non-assembled
USDA257	GCF_024400375.1	7.0	1 chromosome (6.48) and **one plasmid** (19 contigs, non-assembled, **0.56**)

^a^ Available at https://www.ncbi.nlm.nih.gov/datasets/genome/ (accessed on 31 July 2025).

**Table 5 genes-16-01094-t005:** Number of genes related to mobile elements that have been annotated in different rhizobial genomes.

Rhizobial Strain	Number of Genes Related to Mobile Elements ^1^
*R. leguminosarum* SM52	90
*S. americanum* CCGM7	113
*S. kummerowiae* CCBAU71714	172
*S. medicae* WSM419	191
*S. meliloti* 1021	158
*S. meliloti* Rm41	118
*S. sojae* CCBAU05684	194
*S. terangae* CB3126	108
*S. fredii_*CCBAU45436	260
*S. fredii* HH103 (updated)	340
*S. fredii* NGR234	240
*S. fredii* SMH12	352
*S. fredii* USDA192	286
*S. fredii* USDA193	282
*S. fredii* USDA205	356
*S. fredii* USDA257	209

^1^ Search words: transposase, insertion sequence, mobile[_]element, mobile element[_]protein, IS[_], *tnpA*.

**Table 6 genes-16-01094-t006:** Distribution of genes related to mobile elements among the different replicons of *S. fredii* HH103.

Replicon	Number of Genes Related to Mobile Elements	Replicon Size (bp)	Genes Related to Mobile Elements per 10 kb
chromosome	135	4,313,942	0.31
pSfHH103a1	2	24,038	0.83
pSfHH103a2	1	25,081	0.40
pSfHH103b	14	61,874	2.26
pSfHH103c	27	144,081	1.87
pSymA	110	605,378	1.82
pSymB	51	2,099,565	0.24

## Data Availability

All the data generated in this work have been deposited in public databases. All the scripts generated in this work, the annotation, and the nucleotide sequence of *Sinorhizobium fredii* HH103, as well as raw sequencing reads from PacBio and Illumina, are available at NCBI GenBank and the Sequence Read Archives, respectively, and are accessible via the accession numbers listed in Table 1.

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
