# Peer review of "Updated Sequence and Annotation of the Broad Host Range Rhizobial Symbiont Sinorhizobium fredii HH103 Genome"

_genes, 2025, doi:10.3390/genes16091094_

Round 1

Reviewer 1 Report

Comments and Suggestions for Authors

In this study, the authors use gold standard genome assembly approaches combining long and short read (in this case, PacBio and Illumina sequencing respectively)  technologies to produce an improved genome assembly for the rhizobial strain Sinorhizobium fredii HH103. This rhizobial strain is most commonly studied due to its symbiotic association with the crop legume soy bean but is notable for its wide range of potential hosts that includes species forming either determinant or indeterminant nodules.

                  The most significant improvement in this updated genome assembly is the complete circular assembly of the plasmid pSfHH103d, one of two symbiotic megaplasmids found in this species, which was incomplete in the current assembly. A small number of novel genes are also annotated in this updated assembly.

                  The observation that the pSfHH103d plasmid contains two identical operons encoding the nifHDK proteins (which collectively form the nitrogenase complex) is certainly very eye-catching. Aside from this, most of the analysis performed by the authors (phylogenetic tree of Sinorhizobium spp., comparison of numbers of transposable elements etc) have certainly been performed robustly but are not that interesting and would likely produce similar results with previous versions of the genome. Perhaps the most interesting outcome of this work is not really mentioned by the authors; a complete sequence of the megaplasmid containing key symbiotic genes provides much better tools for researchers to probe the functions of these genes and further our understanding of legume-rhizobia symbiosis in a strain that is noteworthy for its broad host range!

Overall, with roughly 10 years elapsed since the most recent update to the genome assembly for this strain, this work represents a timely improvement to the genetic resources available for researchers who work with this strain. 

Minor comments:

[Throughout manuscript]: The nomenclature used to describe the chromosome and each of the five plasmids that comprise to the HH103 genome is rather varied. For instance, the plasmid which receives most attention in this study seems to be alternatively named p_d, pSfHH103d, pSym or pSymA. Instead of using these names interchangeably throughout the manuscript, the authors should settle on one name for each plasmid and stick with it throughout. Given that most readers of this manuscript would likely be from a legume-rhizobium symbiosis background, using the name (e.g. pSymA) which gives most information about that plasmids function might be most sensible.

127: “The largest plasmids, pSfHH103e and pSfHH103a […]” – I am assuming this is a typo and should refer to plasmids ‘e’ and ‘d’ rather than ‘e’ and ‘a’. But this should be corrected urgently as this will cause significant confusion to readers.

138-146: The observation of duplicated structural nitrogenase genes was very interesting and to the best of my knowledge is not common in Sinorhizobium strains and indeed, genetic redundancy is not common in bacterial genomes. Whilst its clearly not in the scope of this work to investigate this much further, the genes are described as being ‘100% identical’ copies. Does this only refer to the coding sequences of these genes? Is it possible that they may have alternative promoters of other regulatory elements? 

Proofreading:

79: Suggest replacement of “The assembly was polished with […]” to “Draft genome assembly improvement was carried out using […]”. This would likely be easier for readers not familiar with genome assembly methods to understand what is going on at this step.

81: Suggest edit: “[…] Circlator v1.5.5 with the fixstart options was used.” to “[…] Circlator v1.5.5 was used with the fixstart options enabled.”

101: Suggest remove “by”.

153: Suggest edit: “[…] previously HH103 described genes […]” to “[…] previously described HH103 genes […]”.

198: Suggest edit “Comparison of the genomes of HH103 and other rhizobia” to “Comparison of the genome of HH103 with other rhizobia” or “Comparison of the genome of HH103 to other rhizobia”.

223: suggest edit “we were interested in analyzing […]” to “we analyzed […]”.

240-243: Currently two consecutive sentences are starting with “In fact […]”; suggest changing one of these.

Author Response

Reviewer 1

In this study, the authors use gold standard genome assembly approaches combining long and short read (in this case, PacBio and Illumina sequencing respectively) technologies to produce an improved genome assembly for the rhizobial strain Sinorhizobium fredii HH103. This rhizobial strain is most commonly studied due to its symbiotic association with the crop legume soybean but is notable for its wide range of potential hosts that includes species forming either determinant or indeterminant nodules.

The most significant improvement in this updated genome assembly is the complete circular assembly of the plasmid pSfHH103d, one of two symbiotic megaplasmids found in this species, which was incomplete in the current assembly. A small number of novel genes are also annotated in this updated assembly.

The observation that the pSfHH103d plasmid contains two identical operons encoding the NifHDK proteins (which collectively form the nitrogenase complex) is certainly very eye-catching. Aside from this, most of the analysis performed by the authors (phylogenetic tree of Sinorhizobium spp., comparison of numbers of transposable elements etc) have certainly been performed robustly but are not that interesting and would likely produce similar results with previous versions of the genome. Perhaps the most interesting outcome of this work is not really mentioned by the authors; a complete sequence of the megaplasmid containing key symbiotic genes provides much better tools for researchers to probe the functions of these genes and further our understanding of legume-rhizobia symbiosis in a strain that is noteworthy for its broad host range!

Overall, with roughly 10 years elapsed since the most recent update to the genome assembly for this strain, this work represents a timely improvement to the genetic resources available for researchers who work with this strain.

Our answer: Thank you very much for your kind comments. Following your suggestion, we have included a sentence (lines 147-150 and 303-305 of the revised version) stating the importance of having a complete sequence of the pSymA megaplasmid).

Minor comments:

[Throughout manuscript]: The nomenclature used to describe the chromosome and each of the five plasmids that comprise to the HH103 genome is rather varied. For instance, the plasmid which receives most attention in this study seems to be alternatively named p_d, pSfHH103d, pSym or pSymA. Instead of using these names interchangeably throughout the manuscript, the authors should settle on one name for each plasmid and stick with it throughout. Given that most readers of this manuscript would likely be from a legume-rhizobium symbiosis background, using the name (e.g. pSymA) which gives most information about that plasmids function might be most sensible.

Our answer: You are right. We have changed this nomenclature throughout the complete manuscript.

127: “The largest plasmids, pSfHH103e and pSfHH103a […]” – I am assuming this is a typo and should refer to plasmids ‘e’ and ‘d’ rather than ‘e’ and ‘a’. But this should be corrected urgently as this will cause significant confusion to readers.

Our answer: You are completely right. This mistake has been corrected.

138-146: The observation of duplicated structural nitrogenase genes was very interesting and to the best of my knowledge is not common in Sinorhizobium strains and indeed, genetic redundancy is not common in bacterial genomes. Whilst its clearly not in the scope of this work to investigate this much further, the genes are described as being ‘100% identical’ copies. Does this only refer to the coding sequences of these genes? Is it possible that they may have alternative promoters of other regulatory elements?

Our answer: Thanks for your observation. In fact, the presence of two copies of nifHDK is common among S. fredii but not in S. meliloti. Regarding the two copies of nifHDK found in HH103, they both present a NifA-box in its 5’ upstream sequence, which suggests a similar regulation, although further studies will be required to elucidate this point. All this information has been included in the revised version of the manuscript (lines 156-166). We have also included a figure that shows the genetic context of these two copies of nifHDK (New Figure 2).

Proofreading:

79: Suggest replacement of “The assembly was polished with […]” to “Draft genome assembly improvement was carried out using […]”. This would likely be easier for readers not familiar with genome assembly methods to understand what is going on at this step.

Done

81: Suggest edit: “[…] Circlator v1.5.5 with the fixstart options was used.” to “[…] Circlator v1.5.5 was used with the fixstart options enabled.”

Done

101: Suggest remove “by”.

Done

153: Suggest edit: “[…] previously HH103 described genes […]” to “[…] previously described HH103 genes […]”.

Done

198: Suggest edit “Comparison of the genomes of HH103 and other rhizobia” to “Comparison of the genome of HH103 with other rhizobia” or “Comparison of the genome of HH103 to other rhizobia”.

Done

223: suggest edit “we were interested in analyzing […]” to “we analyzed […]”.

Done

240-243: Currently two consecutive sentences are starting with “In fact […]”; suggest changing one of these.

Done

Reviewer 2 Report

Comments and Suggestions for Authors

Line 73:
Have you considered using Nanopore?

Line 79:
Just curious, does Pilon have better performance than Medaka? Have you tried it?

Annotation pipeline question:
Which pipeline did you use for annotation?

Lines 99–114:
Maybe put this information under “Data Availability.”

Author Response

Reviewer 2

Our answer: Thank you very much for your kind suggestions.

  • Line 73:Have you considered using Nanopore?

Our answer: We chose PacBio because, like Nanopore, it provides long read lengths but has the advantage of a minor error rate than Nanopore. In any case, thank you for your suggestion.

  • Line 79:Just curious, does Pilon have better performance than Medaka? Have you tried it?

Our answer: We have not tried to polish with Medaka. Medaka is good for Nanopore reads, but Pilon is more accurate than Medaka when we are combining PacBio and Illumina reads.

  • Annotation pipeline question:Which pipeline did you use for annotation?

Our answer: For the annotation we used bedtools and blastn commands. We extract the previous sequences by using the annotation coordinates and then those sequences were searched in the new sequences. To annotate the remaining gaps we used the automatic annotator bakta.

Lines 99–114: Maybe put this information under “Data Availability.”

Our answer: Done.

Reviewer 3 Report

Comments and Suggestions for Authors

The MDPI manuscript ID: genes - 3821739 presents a relevant study that investigates the updated genome assembly of Sinorhizobium fredii HH103 using a combination of PacBio and Illumina sequencing. The authors provide important insights into replicon organisation, the resolution of the symbiotic plasmid, and the abundance of transposable elements linked to genome plasticity. However, a minor revision would be to incorporate more recent references in the introduction to strengthen the contextual framework, as well as to include additional references supporting the section on the sequencing of S. fredii HH103.

Q1

Page 1 – Line 33

You may wish to strengthen the introduction by complementing the classical reference of Pueppke and Broughton (1999) with more recent reviews that highlight the relevance of S. fredii as symbiotic nitrogen-fixing bacteria. For instance, Souza & Pereira (2019) and Kawaka (2022) provide updated perspectives on conventional and unconventional symbionts, thereby situating the importance of S. fredii within a broader and more current context.

Souza LF, Pereira AC. Conventional and unconventional symbiotic nitrogen fixing bacteria associated with legumes. In: Silva J, Rodrigues M, editors. Biological Nitrogen Fixation and Legume Symbiosis. 2nd ed. Cham: Springer; 2019. p.145–78.

Kawaka F. Characterization of symbiotic and nitrogen fixing bacteria. AMB Express. 2022;12:99. doi:10.1186/s13568-022-01441-7

Q2

Page 2 – Line 47

The description of the HH103 genome is well supported by Margaret et al. (2011). To further contextualise the observed plasmid complexity and diversity, I suggest also citing Tian et al. (2012, PNAS), which discusses plasmid variation and lineage-specific adaptations among soybean-nodulating rhizobia.

Tian CF, Zhou YJ, Zhang YM, et al. Comparative genomics of rhizobia nodulating soybean suggests extensive recruitment of lineage-specific genes in adaptations. Proc Natl Acad Sci U S A. 2012;109(22):8629-8634. doi:10.1073/pnas.1120436109

Author Response

The MDPI manuscript ID: genes - 3821739 presents a relevant study that investigates the updated genome assembly of Sinorhizobium fredii HH103 using a combination of PacBio and Illumina sequencing. The authors provide important insights into replicon organisation, the resolution of the symbiotic plasmid, and the abundance of transposable elements linked to genome plasticity. However, a minor revision would be to incorporate more recent references in the introduction to strengthen the contextual framework, as well as to include additional references supporting the section on the sequencing of S. fredii HH103.

Our answer: Thank you very much for your kind comments and suggestions.

Q1, Page 1 – Line 33

You may wish to strengthen the introduction by complementing the classical reference of Pueppke and Broughton (1999) with more recent reviews that highlight the relevance of S. fredii as symbiotic nitrogen-fixing bacteria. For instance, Souza & Pereira (2019) and Kawaka (2022) provide updated perspectives on conventional and unconventional symbionts, thereby situating the importance of S. fredii within a broader and more current context.

Souza LF, Pereira AC. Conventional and unconventional symbiotic nitrogen fixing bacteria associated with legumes. In: Silva J, Rodrigues M, editors. Biological Nitrogen Fixation and Legume Symbiosis. 2nd ed. Cham: Springer; 2019. p.145–78.

Kawaka F. Characterization of symbiotic and nitrogen fixing bacteria. AMB Express. 2022;12:99. doi:10.1186/s13568-022-01441-7

Our answer: Thank you very much for your useful suggestions. Unfortunately, we could not find the first reference (Souza and Pereira 2019), However, we found a referenc with the same title and subject (3.   El Idrissi, M.M.; Kaddouri, K.; Bouhnik, O.; Lamrabet, M.; Alami, S.; Abdelmoumen, H. Conventional and unconventional symbiotic nitrogen fixing bacteria associated with legumes. In: Developments in Applied Microbiology and Biotechnology, Microbial Symbionts; Dharumadurai, D., Ed.; Academic Press, 2023. pp. 75-109. ISBN 9780323993340. https://doi.org/10.1016/B978-0-323-99334-0.00038-4). We have included this reference and Kawaka 2022 in the Introduction section.

Q2

Page 2 – Line 47

The description of the HH103 genome is well supported by Margaret et al. (2011). To further contextualise the observed plasmid complexity and diversity, I suggest also citing Tian et al. (2012, PNAS), which discusses plasmid variation and lineage-specific adaptations among soybean-nodulating rhizobia.

Tian CF, Zhou YJ, Zhang YM, et al. Comparative genomics of rhizobia nodulating soybean suggests extensive recruitment of lineage-specific genes in adaptations. Proc Natl Acad Sci U S A. 2012;109(22):8629-8634. doi:10.1073/pnas.1120436109

Our answer: Thank you very much for this very appropriate suggestion. This reference has been incorporated in the Introduction section.

Reviewer 4 Report

Comments and Suggestions for Authors

Dear Authors,

The topic of development is innovative, providing detailed and up-to-date information about the genome sequence of Sinorhizobium fredii HH103. The species belongs to the group of nitrogen-fixing bacteria colonizing the root system of legumes. It is often used for biological enrichment of soil with nitrogen, reducing the need for chemical fertilizers. The species serves as a model for the study of molecular symbiosis, genetic regulation and horizontal gene transfer. It has been successfully applied for more sustainable agriculture and soil restoration.

The purpose of the study is specifically formulated and corresponds to the results achieved. Modern literature is used, including enough current literary sources. The study is of very high scientific value because it provides complete information about the plasmid pSfHH103d responsible for nitrogen fixation. The genome of HH103 is one of the most complex among rhizobia, with multiple mobile elements and plasmids, including the symbiotic plasmid pSfHH103d. And in the study, it is proven that the nitrogen-fixing ability of the species is due to it.

The methodical part of the study is presented very well using PacBio - long readings and Illumina - high accuracy, which is a good combination for determining complex genomes. Using the Flye, Pilon, Circlator and Bakta instruments, the previous annotation was combined with the newly identified genes. A very good impression is made by the construction of the phylogenetic tree, through which the relationships between the different species are highlighted.

Tables and graphs illustrate very well the results obtained during the study. PacBio and Illumina solved the problem of having many clusters of transposable elements for the assembly of the pSfHH103d plasmid. The studied symbiotic plasmid of Sinorhizobium fredii was completely assembled with two identical sets of nifHDK genes. During the study, 16 new genes were identified. Using the most modern technology, the plasmid pSfHH103d was fully sequenced.  It has been shown that updating the genome sequence involves an increase in the size of the largest replicants (chromosome, pSfHH103d and pSfHH103e). Many transposable elements have been found in the HH103 genome, especially in three plasmids (pSfHH103b, pSfHH103c, and pSfHH103d) that are hypothesized to be associated with nitrogen fixation.

In conclusion, I think that the article is very well illustrated and methodically maintained. The information obtained about the updated sequence and annotation of the HH103 plasmid is the main basis for further research in the field of symbiosis, fertilization, ecology, microbiology.

Author Response

Dear Authors,

The topic of development is innovative, providing detailed and up-to-date information about the genome sequence of Sinorhizobium fredii HH103. The species belongs to the group of nitrogen-fixing bacteria colonizing the root system of legumes. It is often used for biological enrichment of soil with nitrogen, reducing the need for chemical fertilizers. The species serves as a model for the study of molecular symbiosis, genetic regulation and horizontal gene transfer. It has been successfully applied for more sustainable agriculture and soil restoration.

The purpose of the study is specifically formulated and corresponds to the results achieved. Modern literature is used, including enough current literary sources. The study is of very high scientific value because it provides complete information about the plasmid pSfHH103d responsible for nitrogen fixation. The genome of HH103 is one of the most complex among rhizobia, with multiple mobile elements and plasmids, including the symbiotic plasmid pSfHH103d. And in the study, it is proven that the nitrogen-fixing ability of the species is due to it.

The methodical part of the study is presented very well using PacBio - long readings and Illumina - high accuracy, which is a good combination for determining complex genomes. Using the Flye, Pilon, Circlator and Bakta instruments, the previous annotation was combined with the newly identified genes. A very good impression is made by the construction of the phylogenetic tree, through which the relationships between the different species are highlighted.

Tables and graphs illustrate very well the results obtained during the study. PacBio and Illumina solved the problem of having many clusters of transposable elements for the assembly of the pSfHH103d plasmid. The studied symbiotic plasmid of Sinorhizobium fredii was completely assembled with two identical sets of nifHDK genes. During the study, 16 new genes were identified. Using the most modern technology, the plasmid pSfHH103d was fully sequenced.  It has been shown that updating the genome sequence involves an increase in the size of the largest replicants (chromosome, pSfHH103d and pSfHH103e). Many transposable elements have been found in the HH103 genome, especially in three plasmids (pSfHH103b, pSfHH103c, and pSfHH103d) that are hypothesized to be associated with nitrogen fixation.

In conclusion, I think that the article is very well illustrated and methodically maintained. The information obtained about the updated sequence and annotation of the HH103 plasmid is the main basis for further research in the field of symbiosis, fertilization, ecology, microbiology.

Our answer: Thank you very much for your very kind comments. We are really glad that you enjoyed reading this manuscript.
